# Evaluating the Fungal Pathogens' Inhibition Efficiency of Composite Film Combined with Antagonistic Yeasts and Sodium Alginate on Peach

**Xiaolong Du** [1,*], **Shaobin Li** [1], **An Luo** [1], **Xiaoli Yin** [1], **Kai Fan** [1], **Linyun Mou** [2] **and Jianlong Li** [2]

[1] School of Life Science, Yangtze University, Jingzhou 434025, China
[2] School of Life Sciences, Nanjing University, Nanjing 210023, China
[*] Correspondence: duxiaolong@yangtzeu.edu.cn

**Abstract:** To reduce the indiscriminate use of pesticides and extend the postharvest shelf life of peach fruit (*Prunus persica*, cv. Baihua) from southeast China, mainly the microbial antagonism of indigenous yeasts was studied and applied in the construction of composite film. In this study, 14 yeast strains of 9 genera were screened out from the surface of peaches by isolation, purification, cultivation, and identification. Through an experimental analysis of the in vitro inhibition zone and the in vivo colonizing capacity, $1 \times 10^8$ CFU mL$^{-1}$ of *Candida oleophila* sp-ELPY12B and *Cryptococcus laurentii* sp-ELPY15A proved most efficient against the major pathogens and were chosen as candidate fungicides. In combination with Na-alginate film (0.4% glycerin as the plasticizer and 0.1% Tween-80 as the emulsifier), the preservative effects of these composite-treated groups also showed the best antifungal effects, which significantly delayed the postharvest preservation period by about 6–7 d under an ambient temperature of $25 \pm 3\ ^\circ$C and a relative humidity of 50–70%.

**Keywords:** peach (*Prunus persica*); postharvest preservation; antagonistic yeasts; Na-alginate film; antifungal activity

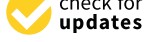



## 1. Introduction

For a long time, chemical fungicides have been used to control peach (*Prunus persica*) postharvest diseases, including Imazalil, Tecto, Carbendazim, thiophanate-methyl, and Nurelle-D505. As environmental pollution becomes more and more severe, the indiscriminate use of these chemical fungicides will not meet the requirements of safety and sanitary conditions, which are increasingly limited by governments. It also further leads to drug-resistance problems in pathogens. Therefore, many areas have been committed to the development of biological prevention and control of antagonistic microorganisms, enzymes, resistance genes, and natural products [1]. Among them, the research on antagonistic yeasts in the control of the postharvest diseases of fruits and vegetables, which started in the 1980s, offers more-generalizable practice and experience. Some of the antagonistic yeasts have started to be patented and commercialized in the laboratory, such as Aspire (*Candida oleophila*, Barcelona, Spain), the Biosave series (*Pseudomonas syringae*, California U.S.A), Shemer (*Metschnikowia fructicola*, Antigua-Barbuda), Serenade (*Bacillus subtilis*, California, U.S.A), Pantovital (*Pantoea agglomerans*, Granada Spain), Boni-protect (*Aureobasidium pullulans*, Berlin Germany), and so on, which have primarily shown good effects. A promising antagonistic yeast requires the following desirable properties: [2] genetic stability, low nutrition demands, good adaptability, and a broad antifungal property. Additionally, it has to be proven harmless to the human body and the host. Thus, the development of a biological control product is a long and costly process requiring continuous input.

In addition to the antagonistic yeasts, the combination of biopolymer edible film could further enhance the total preservation efficiency mainly by changing the surface morphology of fruit and reducing water evaporation and oxygen permeability in the pericarp [3].

As sodium alginate (Na-alginate, which is an anionic natural polymer polysaccharide) has properties of biodegradability, biocompatibility, nontoxicity, adhesion, solubility, and good toughness [4], it can be applied as green packaging material, in this study. Its film form can be easily cleaned with water, which is very suitable for the development of edible films [5].

Currently, the fresh peach fruit (*Prunus persica*, cv. Baihua) from southeast China ripens from July to August with a high yield. The melting-flesh pericarp is liable to decay during harvest, which results in huge economic losses. The shelf life is relatively short, only approximately 2–3 d [6]. To solve the abovementioned problems, this study first carried out detailed research on the isolation, purification, cultivation, and identification of indigenous yeasts on peaches. Subsequently, the candidate antagonistic yeasts of major fungal pathogens were screened out by experiments on their inhibition zone and colonizing capacity. By combining the antagonistic yeasts with Na-alginate, the comprehensive film solutions were prepared, and their preservation effects were evaluated by physiological and organoleptic indicators of weight loss, total soluble solids content, respiration rate, and the decay index under an ambient temperature of 25 ± 3 °C and an RH of 50–70%.

## 2. Materials and Methods

### 2.1. Peach Sampling and Cultivation of Microorganisms

Honey peaches (*Prunus persica*, cv. Baihua) at approximately 80% maturity of similar size and weight (about 7 cm diameter and 200 g), with no mechanical injury or pests, were selected as experimental samples from Fenghuang, Zhangjiagang, Jiangsu, China (31°45′ N, 120°36′ E). To isolate the antagonistic yeasts on the peaches' surface, peaches were suspended in 300 mL of sterile deionized water and fully oscillated on a rotary shaker at 200 rpm (revolutions per minute) for 40 min. Afterward, 60 μL of the above suspension was extracted and evenly coated on YPDA (yeast extract peptone dextrose agar) plates. The medium contained 1% yeast extract, 2% peptone, 2% dextrose (glucose), and 2% agar, separately sterilized at 115 °C for 20 min and then mixed together. It also contained 40 μg mL$^{-1}$ ampicillin and 37.5 μg mL$^{-1}$ methicillin in order to prevent contamination from bacteria. The plates were incubated at 28 °C for 60 h, and the recognized colonies were subsequently purified twice by using the streaking plate method under the same conditions.

Six pathogenic fungal strains, *Aspergillus* (*A. tubingensis*), *Penicillium* (*P. expansum*), *Botrytis* (*B. elliptica*), *Rhizopus* (*R. stolonifer*), *Alternaria* (*A. alternate*), and *Monilinia* (*M. fructicola*), were identified and isolated from decay peaches in our past studies [6]. They were selected as antagonistic objectives in inhibition experiments for antagonistic yeasts, which were stored on PDA (potato dextrose agar) slants. The medium contained 20% potato, 2% dextrose, and 2% agar, separately sterilized at 115 °C for 20 min). It also contained 50 μg mL$^{-1}$ of streptomycin at 4 °C. The growing fungal spores were then gently scraped off for PD-liquid-medium cultivation (400 mL). After having been cultured at appropriate temperatures 26 ± 2 °C for *Aspergillus*, *Penicillium*, *Rhizopus*, *Alternaria*, and *Monilinia* and 20–22 °C for Botrytis for 72 h, the spore suspensions of each strain were collected by filtration (7 layers of cheese cloth, 3 times). By mixing and thorough oscillation, the appropriate dilutions (gradient dilution method, 10–10$^4$ times) were pipetted into a hemocytometer to be counted. They were then diluted to the required concentrations.

### 2.2. Molecular Identification of Antagonistic Yeasts

The yeast strains were identified by sequencing the genes of internal transcribed spacer (ITS) of 5.8S ribosomal RNA (forward primer: 5′-TCCTCCGCTTATTGATATGC-3′; reverse primer: 5′-GGAAGTAAAAGTCGTAACAAGG-3′) and D1/D2 domain (forward primer: 5′-GCATATCAATAAGCGGAGGAAAAG-3′; reverse primer: 5′-GGTCCGTGTTTCAAGACGG-3′) at the 5′ end of the 26S rRNA. The polymerase chain reaction (PCR) was conducted according to White [7] by using universal primers. The products were then sent to Sangon Biological Engineering Technology Co., Ltd. (Shanghai, China) for sequencing, and the sequences were identified by BLASTn (NCBI, https://blast.ncbi.nlm.nih.gov/Blast.cgi,

accessed on 8 October 2022). After being aligned by MEGA 7, related phylogenetic trees were constructed by using the neighbor-joining method (Kimura' 2-parameter model) [8,9] from an evolutionary distance, concluding the type of strains from American Type Culture Collection (ATCC), Agricultural Research Service Culture Collection (NRRL), and Centraalbureau voor Schimmelcultures (CBS). The estimation of statistics was measured by using the bootstrap method (1000 replications) [10]. The referential standard strains were represented as genus and species by binomial nomenclature and accession numbers from GenBank (https://www.ncbi.nlm.nih.gov/genbank/, accessed on 28 October 2022).

### 2.3. In Vitro Antifungal Test of Antagonistic Yeasts

To preliminarily evaluate the antifungal effect of the screened yeasts, the in vitro inhibition zone experiments were conducted by using the beating holes method, as follows: 100 μL of pathogenic fungal suspensions at $1 \times 10^6$ CFU mL$^{-1}$ were separately prepared and evenly smeared on Mueller-Hinton agar mediums (0.6% beef extract, 2% peptone, 0.5% sodium chloride, pH = 7.4, sterilized at 121 °C for 20 min). The agar of the mediums was then perforated by sterilized punch or steel pipe to make holes (6 mm) to contain the yeast suspensions (60 μL, $1 \times 10^8$ CFU mL$^{-1}$) for the test. Referring to the Standard of Clinical and Laboratory Standards Institute (CLSI) of 2022, the wounds filled with sterile water were defined as negative controls, and amphotericin B was defined as a positive control. After being cultured at an ambient temperature of $25 \pm 3$ °C and an RH of 50%–70% for 48 h, the diameters of the inhibition zones were determined by crossing measurement with vernier calipers [11]. The total relative inhibitory rate (TRI) was calculated by using Equation (1):

$$TRI = \left( \bar{d}_{ex} - \bar{d}_{nc} \right) / \left( \bar{d}_{pc} - \bar{d}_{nc} \right) \times 100\% \tag{1}$$

where *TRI* is the total relative inhibitory rate and $\bar{d}_{pc}$, $\bar{d}_{nc}$, and $\bar{d}_{ex}$ represent the averaged inhibition zone diameters under the stress of six pathogens from the positive control (amphotericin B), negative control (sterile water), and experimental groups, respectively. Each yeast treatment was represented by 2 plates, and each plate was measured 3 times to obtain an average.

### 2.4. Colonizing Capacity of Candidate Antagonistic Yeasts

After in vitro antifungal test, the yeasts that have a remarkable antagonistic effect were selected for a subsequently colonizing capacity test and preservative effect analysis on peaches. The experiments were conducted mainly according to Zhang et al. [12], with appropriate modifications. Peaches were punctured with 6 equally distributed wounds (6 mm diameter and 3 mm deep) by sterilized steel pipe on surface to contain 40 μL each of pathogenic fungal suspension applied in this study at $1 \times 10^6$ CFU mL$^{-1}$. After 2 h of absorbing and drying, yeast suspensions ($1 \times 10^8$ CFU mL$^{-1}$) were applied into each wound (15 μL) and sprayed to cover the whole surface, including applying the sterile water as a control check (CK).

To estimate the colonization ability and growth dynamic of the antagonistic yeasts, 2 g pulp tissues (0–5 mm from junction area) of the CK group over the next 10 d were cut off in sequence and grounded into appropriate dilutions with 30 mL of 0.7% saline loading. The yeasts from these tissues were separately recovered by spreading 50 μL of each dilution on YPDA and culturing them at an ambient temperature of $25 \pm 3$ °C and RH of 50%–70% for 36 h. The colonies on plates were then counted and calculated by using the gradient dilution method with a hemocytometer, and the population densities of the antagonistic yeasts were described as log$_{10}$ CFU per wound.

### 2.5. Antagonistic Yeast Preservative Effect Analysis on Fungi-Inoculated Peach

The in vivo preservative effect of antagonistic yeasts were also evaluated. After the inoculation of fungi and yeasts, related quality attributes of weight loss, respiration rate, soluble solids content, and the decay index were measured every 2 d during 14 d of storage

under the same circumstances, with minor modifications [6]. In decay index measurement, the statistics of disease speckle coverage should ignore the area of initial punched wounds.

### 2.6. Evaluating the Fungal Pathogens' Inhibition Efficiency for Composite Films

For further application in production, Na-alginate was applied with antagonistic yeast solution to form edible films to enhance the preservative effect. The edible film solutions were mainly prepared according to the procedure used by Foteini et al. [13], with modifications. First, 1 g of Na-alginate was added and sufficiently dissolved in 100 mL of prewarmed (50–70 °C) sterile water by using a dispersion homogenizer (3000 rpm). Following gradual addition of 0.4% glycerin (*w/v*, hereinafter inclusive) as plasticizer and 0.1% Tween-80 as emulsifier, the coating solution was then fully vibrated by ultrasonic cleaner for 20 min. After degassing with vacuum pump and cooling down to $25 \pm 3$ °C, the candidate yeast isolate was added with agitation, to yield final concentration of $1 \times 10^8$ CFU mL$^{-1}$, and the pH of the mixed solution was adjusted to 6–7 by using 0.1 mol L$^{-1}$ acetic acid. To verify the antifungal effects of yeasts with edible films, the relative solutions were applied on fungi-inoculated peaches. All experiments were conducted at an ambient temperature of $25 \pm 3$ °C and RH of 50–70%, and details of all groups are demonstrated in Table 1. Further, 7 fruits were regarded as a replicate, and 3 replicates were measured per treatment to obtain an average.

**Table 1.** Treatments of different groups.

| Group Name | Treatments | Instructions |
|---|---|---|
| Control Check (CK) | Sterile Water | • Fungi-inoculated peach without yeast treatment |
| Group 1 | Na-alginate film | • The composite Na-alginate film was combined with 0.4% glycerin as plasticizer and 0.1% Tween-80 as emulsifier |
| Group 2 | *Cryptococcus laurentii* | • $1 \times 10^8$ CFU mL$^{-1}$ of yeast suspension was applied |
| Group 3 | *Candida oleophila* | |
| Group 4 | Na-alginate film composite by *Cryptococcus laurentii* | • The composite Na-alginate coating solutions were mixed with $1 \times 10^8$ CFU mL$^{-1}$ of yeast |
| Group 5 | Na-alginate film composite by *Candida oleophila* | |

### 2.7. Statistical Data Processing

The experimental results were analyzed by SPSS 22.0 for Windows. Analysis of variance (ANOVA) and Duncan's multiple ranges at confidence interval of $p < 0.05$ were determined to be significantly different among different treatments. The related analysis graphics were created by OriginPro 2020.

## 3. Results

### 3.1. Isolation and Identification of Yeasts on Peaches

In this study, 14 yeast strains from 9 genera were separately isolated, and most of them were identified and confirmed as currently recognized yeast species through ITS and D1/D2 domain sequence alignment with the GenBank database. They are *Metschnikowia* (*M. citriensis, M. zizyphicola*), *Pichia* (*P. fermentans, P. anomala*), *Meyerozyma guilliermondii*, *Candida* (*C. glabrata, C. inconspicua, C. oleophila*), *Torulaspora delbrueckii, Clavispora lusitaniae, Rhodotorula glutinis, Cryptococcus* (*C. laurentii, C. flavescens*), and *Sporobolomyces roseus*, in which three strains were found to have significantly different gene sequences from the corresponding known species. To further identify the three strains, relative phylogenetic trees were constructed by D1/D2 domain sequence comparisons; they are illustrated in Figure 1.

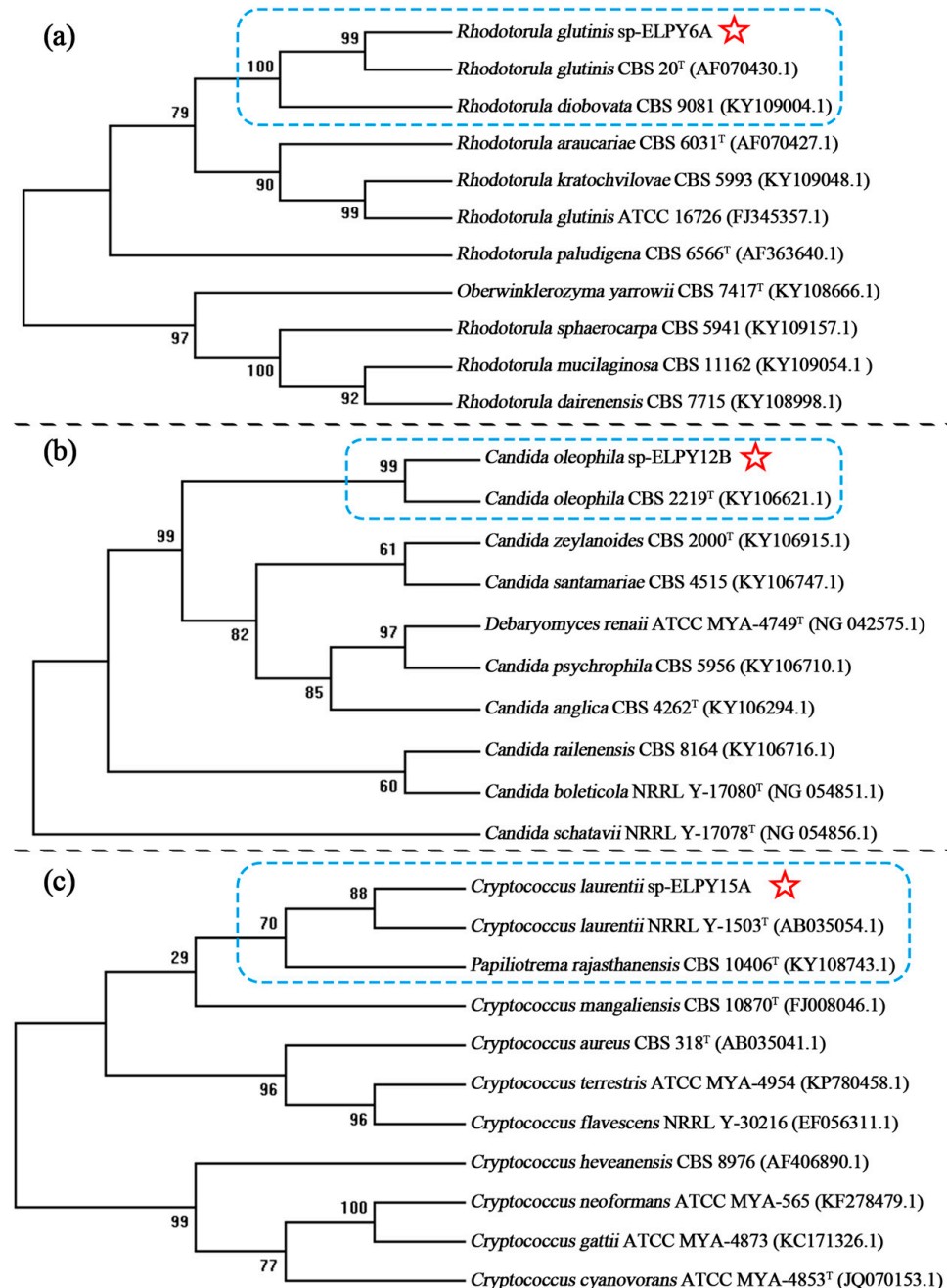

**Figure 1.** The phylogenetic trees of 26S rRNA D1/D2 domain sequences from the three strains. The types of strains from ATCC, NRRL, and CBS are marked with a superscript T letter. (**a**) is the phylogenetic tree of *Rhodotorula glutinis* sp-ELPY6A, (**b**) is the phylogenetic tree of *Candida oleophila* sp-ELPY12B, (**c**) is the phylogenetic tree of *Cryptococcus laurentii* sp-ELPY15A. red star is the candidate yeast with genetic differences.

As shown in Figure 1a, the new isolated strain belongs to the clade of *Rhodotorula*, according to an overall comparison. Within closely related taxa, it is most similar to *R. glutinis* CBS 20T (AF070430.1) and *R. diobovata* CBS 9081 (KY109004.1). As the strain has lower sequence coverage with *R. diobovata*, it was eventually identified as *Rhodotorula glutinis* sp-ELPY6A (assigned by specific laboratory number). From Figure 1b, the unidentified strain has significant sequence similarities and a high bootstrap value with *Candida oleophila* CBS 2219T (KY106621.1), so it was named *Candida oleophila* sp-ELPY12B. In Figure 1c, the unidentified strain proved to have more similarity to *Cryptococcus laurentii* NRRL Y-1503T

(AB035054.1) than to *Papiliotrema rajasthanensis* CBS 10406T (KY108743.1), and the bootstrap value is acceptable. The strain was identified as *Cryptococcus laurentii* sp-ELPY15A. Past research has established the standard for identification: more than a 1% sequence difference from 26S rDNA D1/D2 domain in yeast strains was defined as a different species [14]. Therefore, the strains of *Rhodotorula glutinis* sp-ELPY6A (2.49%, 14 out of 561 nucleotides), *Candida oleophila* sp-ELPY12B (2.03%, 12 out of 589 nucleotides), and *Cryptococcus laurentii* sp-ELPY15A (1.76%, 10 out of 568 nucleotides) are confirmed as new species of the corresponding genus.

### 3.2. In Vitro Antifungal Test for Screening the Isolated Yeasts

Through isolation, cultivation, and sequence identification, in vitro antifungal tests on all 14 yeasts were conducted by using the inhibition zone method, the measuring data were collected, and the TRI values appear in Table 2. According to the CLSI 2021 standard, the test yeasts that have inhibition zone diameters over 15 mm are considered to be extremely susceptive to the target pathogens. As a common efficient polyene fungicide, amphotericin B showed a high degree of fungistatic activity, and the inhibition zone is obviously lager than other groups, so it is well suited for application as an evaluating reference in this experiment. Overall, the top eight yeasts (from number 3 to number 10) exhibited good antagonistic effects in that the inhibition zone results for most fungi are acceptable (10–15 mm is considered to be moderately susceptible), and the TRI values are relatively higher (over 40%) than those of the remaining six stains. They have almost half the antifungal properties as amphotericin B. However, *Sporobolomyces roseus*, *Metschnikowia citriensis*, and *Metschnikowia zizyphicola* are not so effective at inhibiting *M. fructicola* (6–10 mm is considered to be resistant). Thus, *Cryptococcus laurentii*, *Candida oleophila*, *Meyerozyma guilliermondii*, *Rhodotorula glutinis*, and *Cryptococcus flavescens* were chosen as the primary candidates for the following experiments. On the other hand, it is obvious to find that yeasts from the same genus do not necessarily show similar results in antagonistic efficiency [15], such as the three strains of *Candida* in this research.

### 3.3. Colonizing Capacity of Candidate Antagonistic Yeasts on Peaches

Under the ambient temperature of 25 ± 3 °C and the RH of 50–70%, the growth conditions of candidate yeasts on peaches were evaluated every day over 10 d, and these evaluations are illustrated in Figure 2. After a short adaptation of the lag phase, it is worth noting that all the yeasts had rapid proliferations in the first few days of the logarithmic phase [16]. The populations of *Cryptococcus laurentii*, *Candida oleophila*, *Meyerozyma guilliermondii*, *Rhodotorula glutinis*, and *Cryptococcus flavescens* reached their corresponding maximum values at 5 d, 8 d, 3 d, 9 d, and 6 d; they were $3.98 \times 10^8$ CFU per wound, $1.35 \times 10^8$ CFU per wound, $7.59 \times 10^7$ CFU per wound, $4.37 \times 10^7$ CFU per wound, and $1.82 \times 10^8$ CFU per wound, respectively. The yeast densities then basically remained at the same level or declined slightly for the strains; growth gradually stepped into the stationary phase and the decline phase. Although all the strains had problems of growth stagnation after 5–8 d, referring to the trend lines, *Cryptococcus laurentii*, *Candida oleophila*, and *Cryptococcus flavescens* still maintained rather-high-density levels that were over $1 \times 10^8$ CFU per wound, and they were confirmed to have better colonizing capacities than *Rhodotorula glutinis* and *Meyerozyma guilliermondii*. Therefore, the three strains were considered as proper antagonistic strains that can be applied in the preparation of preservative productions. In addition, the colonization capacity of yeast on fruit was also affected by growth or postharvest conditions, such as temperature, humidity, illumination, air composition, and host species [17]. Thus, this screening result was mainly determined by which peach (*Prunus persica*, cv. Baihua) was used in this experiment.

**Table 2.** The results of inhibition zone with standard deviations.

| No. | The Tested Items | The Diameter of the Inhibition Zone (mm) | | | | | | Total Relative Inhibitory Rate (%) |
|---|---|---|---|---|---|---|---|---|
| | | *A. tubingensis* | *P. expansum* | *B. elliptica* | *R. stolonifer* | *A. alternate* | *M. fructicola* | |
| 1 | Sterile Water ($d_{nc}$) | 6.00 ± 0.00 g | 6.00 ± 0.00 f | 6.00 ± 0.00 f | 6.00 ± 0.00 g | 6.00 ± 0.00 e | 6.00 ± 0.00 g | N/A[+] |
| 2 | Amphotericin B ($d_{pc}$) | 20.53 ± 2.12 a | 18.76 ± 1.56 a | 19.88 ± 1.44 a | 18.94 ± 1.76 a | 19.07 ± 2.11 a | 19.85 ± 2.69 a | N/A |
| 3 | *Cryptococcus laurentii* | 12.73 ± 1.25 bcd | 16.38 ± 1.37 ab | 15.04 ± 1.66 b | 11.15 ± 1.16 def | 14.42 ± 1.58 b | 14.93 ± 1.36 b | 60.04 |
| 4 | *Candida oleophila* | 13.88 ± 1.43 b | 14.62 ± 2.59 c | 12.97 ± 1.79 bcde | 12.64 ± 1.52 bc | 12.58 ± 2.39 bc | 13.35 ± 1.48 b | 54.35 |
| 5 | *Meyerozyma guilliermondii* | 12.48 ± 0.85 bcd | 13.39 ± 1.93 c | 13.25 ± 1.64 bcd | 14.23 ± 0.68 b | 12.36 ± 2.85 bc | 10.70 ± 1.14 bcd | 49.87 |
| 6 | *Sporobolomyces roseus* | 11.52 ± 1.29 cd | 13.83 ± 2.04 bc | 13.75 ± 2.15 bc | 14.22 ± 1.77 b | 13.11 ± 2.53 bc | 9.54 ± 1.02 def | 49.32 |
| 7 | *Metschnikowia citriensis* | 13.65 ± 2.01 b | 13.96 ± 1.83 bc | 10.63 ± 1.00 de | 12.94 ± 2.21 bc | 13.75 ± 2.22 bc | 8.58 ± 1.31 defg | 46.29 |
| 8 | *Rhodotorula glutinis* | 10.63 ± 1.13 de | 11.96 ± 1.38 cd | 13.67 ± 1.68 bc | 13.56 ± 1.67 bc | 12.18 ± 1.37 bc | 10.97 ± 2.01 bcd | 45.62 |
| 9 | *Cryptococcus flavescens* | 10.59 ± 0.96 cd | 12.77 ± 1.76 c | 11.86 ± 1.47 cde | 12.02 ± 1.73 bcd | 11.35 ± 1.33 bc | 12.54 ± 2.37 bc | 43.35 |
| 10 | *Metschnikowia zizyphicola* | 12.74 ± 1.48 bc | 14.11 ± 1.65 bc | 11.42 ± 1.59 cde | 13.07 ± 0.91 bc | 12.24 ± 2.55 bc | 7.35 ± 0.82 fg | 43.11 |
| 11 | *Pichia anomala* | 8.24 ± 0.76 f | 12.42 ± 1.79 c | 10.27 ± 1.06 e | 12.51 ± 1.16 bc | 7.63 ± 1.19 de | 13.46 ± 1.39 b | 35.21 |
| 12 | *Candida glabrata* | 9.11 ± 0.47 ef | 9.20 ± 1.06 e | 11.89 ± 1.79 cde | 11.34 ± 0.88 cde | 10.61 ± 0.95 cd | 8.34 ± 0.73 efg | 30.22 |
| 13 | *Candida inconspicua* | 7.86 ± 0.43 fg | 8.78 ± 0.95 e | 10.56 ± 2.35 de | 8.29 ± 1.26 f | 11.57 ± 1.35 bc | 7.43 ± 0.66 fg | 22.82 |
| 14 | *Pichia fermentans* | 6.00 ± 0.00 g | 9.47 ± 0.84 de | 10.56 ± 1.75 de | 9.68 ± 1.14 ef | 7.93 ± 0.82 de | 10.34 ± 1.39 cde | 22.19 |
| 15 | *Torulaspora delbrueckii* | 6.00 ± 0.00 g | 6.00 ± 0.00 f | 7.49 ± 0.59 f | 6.00 ± 0.00 g | 12.43 ± 2.77 bc | 8.82 ± 0.98 def | 13.25 |
| 16 | *Clavispora lusitaniae* | 6.00 ± 0.00 g | 7.36 ± 0.99 ef | 6.00 ± 0.00 f | 9.81 ± 0.77 def | 6.00 ± 0.00 e | 7.59 ± 1.17 fg | 8.34 |

Note: N/A[+]: not applicable. Detailed information on the species is provided in the text above. The data in the table are shown as the mean ± SD, and the different normal letters in the same column indicate significant differences among treatments at the $p = 0.05$ level.

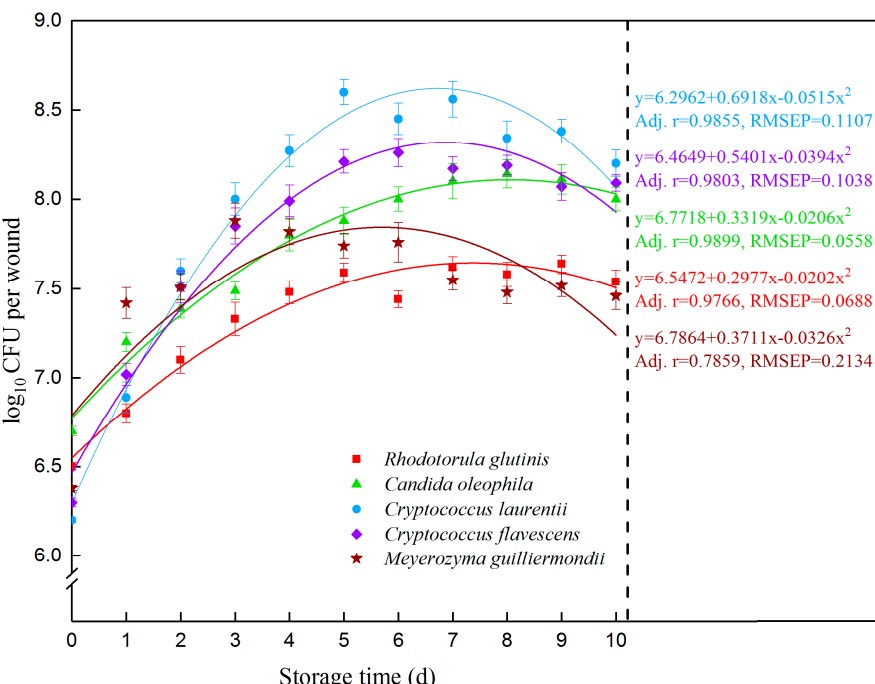

$y=6.2962+0.6918x-0.0515x^2$
Adj. r=0.9855, RMSEP=0.1107

$y=6.4649+0.5401x-0.0394x^2$
Adj. r=0.9803, RMSEP=0.1038

$y=6.7718+0.3319x-0.0206x^2$
Adj. r=0.9899, RMSEP=0.0558

$y=6.5472+0.2977x-0.0202x^2$
Adj. r=0.9766, RMSEP=0.0688

$y=6.7864+0.3711x-0.0326x^2$
Adj. r=0.7859, RMSEP=0.2134

■ *Rhodotorula glutinis*
▲ *Candida oleophila*
● *Cryptococcus laurentii*
◆ *Cryptococcus flavescens*
★ *Meyerozyma guilliermondii*

**Figure 2.** Population dynamics of five candidate yeasts. The concentrations are shown as mean ± SD, and the trend lines are constructed by polynomial regression.

### 3.4. In Vivo Preservative Effects of Candidate Antagonistic Yeasts on Fungi-Inoculated Peaches

To determine the in vitro antifungal effect on and the colonizing capacity of film, *Candida oleophila* sp-ELPY12B and *Cryptococcus laurentii* sp-ELPY15A were finally chosen as microbiological additives in combination with a Na-alginate film solution (0.4% glycerin as plasticizer and 0.1% Tween-80 as emulsifier) for an in vivo preservative effect analysis, in practice. After the quality attributes of weight loss (WL), respiration rate (RR), soluble solid content (SSC), and the decay index (DI) had been evaluated every 2 d during storage, the relative statistical graph was illustrated which is shown in Figure 3.

Past studies have shown that most of the postharvest WL is caused mainly by water loss in transpiration, while the rest is caused by respiration in nutrition consumption [18]. In this study, peaches from different groups suffered a total WL ranging from 16% to 25% after 14 d of storage. From the data shown in Figure 3a, all the curves have initial upward trends and then decrease and remain relatively flat near 2% in general. During physiological postripeness, especially the untreated peaches from the CK group most quickly lost weight by 5–7%, with high respiration activity (Figure 3b), and thereafter, they maintained the lowest weight-loss rate after 6 d. The yeast-suspension-treated groups (group 2 and group 3) still cannot restrain the initial rapid WL compared with the film-treated groups (group 1, group 4, and group 5). All these above phenomena suggested that the edible Na-alginate film on the surface of peaches can isolate themselves from the external environment to a certain extent to effectively reduce the water transpiration, which could greatly reduce the WL in the early stages of harvesting. Although the subsequent WL variability remained low, many peaches lost considerable weight in the first few days, leading to poor quality. For peaches, as a kind of juicy fruit, when the WL is greater than 8%, the peel of the peach pulp begins to shrink, thus affecting the flavor and appearance. By summing each measurement, the total WL of peaches from group CK, group 1, group 2, and group 3 reached the critical value on 3 d, 6 d, 4 d, and 5 d, respectively. In contrast, the acceptable period of peaches from group 4 and group 5 can be deferred to 10 d in storage.

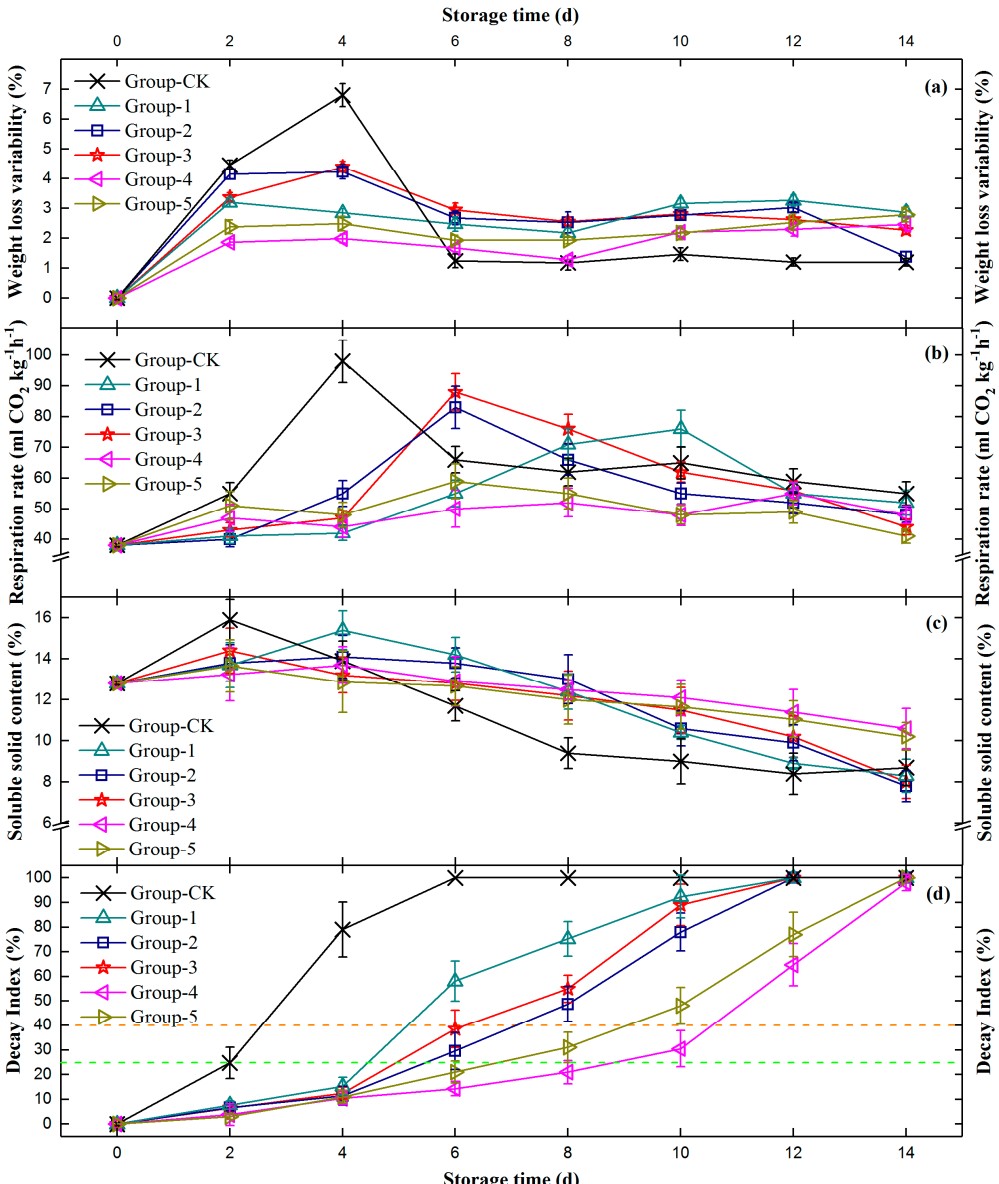

**Figure 3.** Physiological and organoleptic variations from different treatment groups. (**a**) WL, %; (**b**) RR, mL $CO_2$ $kg^{-1}$ $h^{-1}$; (**c**) SSC, %; (**d**) DI, %. Detailed information is provided in Table 1.

　　As shown in Figure 3b, the respiratory peaks of group CK, group 1, group 2, and group 3 were found, and they reached the maximum values of 98 mL $CO_2$ $kg^{-1}$ $h^{-1}$, 76 mL $CO_2$ $kg^{-1}$ $h^{-1}$, 83 mL $CO_2$ $kg^{-1}$ $h^{-1}$, and 88 mL $CO_2$ $kg^{-1}$ $h^{-1}$ on 4 d, 10 d, 6 d, and 6 d, respectively. The approach of the respiration climacteric of the coated peaches from group 1 were delayed by 6 d compared with those of the bare peaches form group CK. It is worth noting that the simple application of antagonistic yeast seems to have little effect on respiratory inhibition, according to a comparison of group 2 and group 3 with group CK. Furthermore, by combining the antagonistic yeasts and the Na-alginate film, the RR of the peaches in group 4 and group 5 was greatly inhibited and remained below 60 $CO_2$ $kg^{-1}$ $h^{-1}$ for the duration, which was mainly due to air separation provided by the film and the inhibition of fungi. According to the data in Figure 3a, changes in respiratory intensity had a relatively small effect on the WL variability, probably because most WL was caused by water transpiration.

　　According to the results shown in Figure 3c, the overall trend of SSC steadily decreases, with the exception of a small initial increase. This is mainly because the peach samples

used in the experiment were not fully ripened at the beginning (0 d), and the SSC gradually increased to the maximum value, along with the hydrolysis of polysaccharides [19]. The untreated peaches from group CK showed a rapid decline in SSC, while the declines in other treatment groups were more gradual, especially in group 4 and group 5. The SSC of the untreated peaches from group CK rapidly decreased, while that of the other treatment groups more slowly decreased, especially group 4 and group 5. In the balance of polysaccharide hydrolysis and SSC consumption, the curve of group 4 and group 5 did not even show an initial increase but remained stable at around 12%.

From the results in Figure 3d, it is obvious that the DI of group CK rapidly increased compared with the other groups and reached 100% by only 5 d. The application of antagonistic yeasts efficiently reduced the DI according to a comparison of the results from group CK with those of group 2 and group 3. In addition, the isolation effect of Na-alginate film (group 1) on the surface of peaches also inhibited the contact of the external fungi to some extent, thus reducing the DI. By combining the antagonistic yeasts and the Na-alginate film, the peaches from group 4 and group 5 remained as fresh as possible in first 7 d and 9 d and were still acceptable (DI < 40%) before 9 d and 10 d, respectively. Compared with the control group, the effective preservation period was extended by about 6–7 d.

## 4. Discussion

Overall, the application of edible Na-alginate film reduces the occurrence of rot by isolating the outside air, compared with naked peaches. At the same time, the film avoided oxidation and respiration to a certain extent. The peach is a fruit that has typical respiration climacteric and short after-ripening periods. Therefore, the RR stands for the metabolism intensity for the most part, and it is usually associated with other indicators, such as firmness, soluble sugar content, and electrical conductivity [20]. The peach applied in this study is a melting-flesh species, which is more perishable and can more rapidly reach its respiratory peak than common species (about 5–6 d) [21]. Thus, the suppression of respiration is especially necessary. The SSC defines mainly the total soluble saccharides that were largely consumed as a respiratory substrate during postharvest, such as sucrose, glucose, and fructose. As the species *Baihua* possesses strong fragrant flavor with high sweetness, the maximum value of SSC could reach up to 16%, which is relatively higher than those of general varieties [22]. In practice, the SSC of well-preserved peaches from these two groups were found to recover to the high level again in a short period without the edible wrapping film, a result that leads to speculation that the physiological activity has changed, which needs further investigation. DI is a direct and useful organoleptic parameter for assessing the comprehensive disease severities of peaches in observations. According to past research experiences [6], a peach with a DI below 25% is considered as having a good preservative status; having a DI being between 25% and 40% is acceptable; and a DI is over 40% is considered inedible. Combined with the previous analysis, the above change in physiological parameters (WL, RR and SSC) will become relatively flat after the peaches have become completely rotten, which is mainly because the fruit itself loses its physiological activity, giving rise to cell membrane rupture and cell disassembly, and then the fungi take over the following metabolism dynamics [23]. The composite biological preservatives prepared by combining antagonistic yeasts with Na-alginate film are more environmentally friendly and easily obtained. The related middle-scale experimental applications of treatments have been conducted in producing areas of Jiangsu province, and they obtained good results.

## 5. Conclusions

By isolating and culturing the yeasts on the surface of peaches from Zhangjiagang, in southeast China, 14 yeast strains of 9 genera were screened out. They are *Metschnikowia* (*M. citriensis*, *M. zizyphicola*), *Pichia* (*P. fermentans*, *P. anomala*), *Meyerozyma guilliermondii*, *Candida* (*C. glabrata*, *C. inconspicua*, *C. oleophila*), *Torulaspora delbrueckii*, *Clavispora lusitaniae*, *Rhodotorula glutinis*, *Cryptococcus* (*C. laurentii*, *C. flavescens*), and *Sporobolomyces roseus*.

Among them, three strains were considered as new species with a significantly different D1/D2 domain sequence, according to a phylogenetic analysis. They were respectively named *Rhodotorula glutinis* sp-ELPY6A, *Candida oleophila* sp-ELPY12B, and *Cryptococcus laurentii* sp-ELPY15A. According to the results of in vitro antifungal tests against six kinds of fungi (*Aspergillus tubingensis*, *Penicillium expansum*, *Botrytis elliptica*, *Rhizopus stolonifer*, *Alternaria alternate*, and *Monilinia fructicola*) and the colonizing capacity tests, $1 \times 10^8$ CFU mL$^{-1}$ of *Candida oleophila* sp-ELPY12B and *Cryptococcus laurentii* sp-ELPY15A proved to be the optimal concentration. Combined with Na-alginate (0.4% glycerin as the plasticizer and 0.1% Tween-80 as the emulsifier), the preservative effects of these composite-treated groups showed the best antifungal effects, which significantly delayed the postharvest preservation period by about 6–7 d under an ambient temperature of $25 \pm 3$ °C and a relative humidity of 50–70%.

**Author Contributions:** Conceptualization, X.D. and J.L.; methodology, X.D. and S.L.; software, A.L.; validation, K.F. and L.M.; formal analysis, X.D.; investigation, X.D.; resources, X.D. and J.L.; data curation, X.Y.; writing—original draft preparation, X.D.; writing—review and editing, X.D. All authors have read and agreed to the published version of the manuscript.

**Funding:** This research was funded by Research of key biotechnology of peach antisepsis and fresh-keeping paper products (the Suzhou Science and Technology Project of China, Grant No.: SNG201447), and the nutrition physiology, decay mechanism, and comprehensive preservation research of Fenghuang honey peach (The major science and technology support plan of Zhangjiagang City, Grant No.: ZKN1002).

**Institutional Review Board Statement:** Not applicable.

**Informed Consent Statement:** Not applicable.

**Data Availability Statement:** Not applicable.

**Acknowledgments:** I acknowledge the assistance of the Fenghuang honey peach processing factory in Zhangjiagang City for providing experimental sites and materials.

**Conflicts of Interest:** The authors declare no conflict of interest.

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
