# Peer review of "Evaluating the Fungal Pathogens’ Inhibition Efficiency of Composite Film Combined with Antagonistic Yeasts and Sodium Alginate on Peach"

_coatings, doi:10.3390/coatings13020417_

Round 1

Reviewer 1 Report

ABTRACT

The abstract contains too long sentence. Clarify by shortening the sentences with a defined object each

It should say: Prunus persica, cv. Baihua.

Reviewer 2 Report

A review report on manuscript titled:

Evaluating the fungal pathogens inhibition efficiency of composite film combined with antagonistic yeasts and sodium alginate on peach

General overview:

The review discussed one of the important topics about using of evaluating the fungal pathogens inhibition efficiency of composite film combined with antagonistic yeasts and sodium alginate on peach. A promising data were provided that reduction of the indiscriminate use of pesticides and extend the postharvest shelf life of peach fruit (cv. Baihua) from southeast China, microbial antagonism of indigenous yeasts was  mainly studied and applied in construction of composite film.

-Since the frame work of the review is promising, it must be improved in its overall quality many typos errors were provided and sentences without verbs. The rigors revision is important.

-The presentation of the manuscript must be much improved. The Introduction and methods should be better described. The discussion is scientifically poor and conclusion section must be provided.

Abstract:

Line 11,12 : Revise the grammar, correction again.

Line 14: Revise and correct; the sentence is long.

Introduction:

Line 25,26; correct and revise the grammar

There are some writing mistake in the format should be correct.

It is the authours’ responsibility to maintain the consistence of reference format.

L40-44 should be revised and rewritten.

L65-68. By the way, the descriptions are not well prepared.

Reviewer 3 Report

It is worth considering publishing the manuscript titled "Evaluating the Fungal Pathogens Inhibition Efficiency of Composite Film Combined with Antagonistic Yeasts and Sodium Alginate on Peach" in its current form since it is intriguing. I advise authors to provide a few of the study's major findings under the title Conclusions.

Reviewer 4 Report

 For a long time, the main methods to control peach (Prunus persica) postharvest diseases are the use of chemical fungicides. It's melting-flesh pericarp is liable to decay during harvest which resulted in huge economic losses, and the shelf life is relatively short, only approximately 2 - 3 d. To solve the above-mentioned problem, this study firstly carried out detailed research on isolation, purification, cultivation and identification of indigenous yeasts on peach. Subsequently, the candidate antagonistic yeasts against major fungal pathogens were screened out by the experiments of inhibition zone and colonizing capacity. To reduce the indiscriminate use of pesticides and extend the postharvest shelf life of peach fruit (cv. Baihua) from southeast China, microbial antagonism of indigenous yeasts was mainly studied and applied in construction of composite film. On this basis, Na-alginate was added in yeast suspension to prepare the film solution for further improvement in practical application, and the comprehensive preservation effects were evaluated by physiological and organoleptic indicators of weight loss, total soluble solids content, respiration rate and decay index under ambient temperature of 25 ± 3 °C and RH of 50 - 70 %.

The authors  found that the SSC of well-preserved peaches would recover to the high level again in short period after washed off the edible wrapping film, speculating that the physiological activity has changed certain degree, which needs further investigation. DI is a direct and useful organoleptic parameter for assessing the comprehensive disease severities of peaches in visual. The composite biological preservatives prepared by combining antagonistic yeasts with Na-alginate film are more environmental friendly and easily obtained. Related middle-scale experimental applications of upon treatments have been conducted in producing areas of Jiangsu province and obtained good results.

The work is well done but I have some remarks:

-  The fig 1, 2, 3 caption must be better arranged

- the authors should edite the references style

- a moderate English changes  required

Round 2

Reviewer 2 Report

The revised manuscript has been improved and the authors have been addressed most of the comments I raised previously